# Do *Malassezia* yeasts colonize the guts of people living with HIV?

Abdourahim Abdillah,[1,2] Isabelle Ravaux,[3] Saadia Mokhtari[3] and Stéphane Ranque[1,2]*

**1.** Aix-Marseille Université, AP-HM, SSA, RITMES, Marseille, France, **2.** IHU Méditerranée Infection, Marseille, France, **3.** Hôpital de Jour, Pôle des Maladies Infectieuses et Tropicales, AP-HM, CHU La Timone, Marseille, France

* stephane.ranque@univ-amu.fr

## Abstract

*Malassezia* yeasts are commensals of human skin. In contrast to culture-based studies, metagenomic studies have detected abundant *Malassezia* reads in the gut, especially in patients living with HIV. Whether *Malassezia* colonizes and persists in the gut remains an open question. This study aimed to describe the influence of HIV-associated immunodeficiency on gut colonization by *Malassezia* and to assess whether *Malassezia* are alive. Stool samples were prospectively collected over one–five visits from ten controls and 23 patients living with HIV (10 had CD4 < 200/ mm$^3$ and 13 had CD4 > 500/mm$^3$). Each sample was cultured and subjected to *Malassezia* viability PCR and both fungal and bacterial metabarcoding. Abundant *M. furfur* colonies were cultured from an HIV-immunocompromised patient. *M. furfur* and *M. globosa* were isolated in very low quantities from healthy volunteers. Viability *Malassezia*-specific qPCR was positive in three HIV-immunocompromised patients. Metagenomic analyses showed that *Malassezia* reads were significantly more abundant in immunocompromised patients living with HIV and erratic over time in all participants. Our findings emphasise that *Malassezia* are rarely cultured from human stool samples, despite the use of specific culture media. Although HIV-related immunosuppression appears to be associated with the presence of *Malassezia*, these yeasts do not persist and colonise the gut, even in immunocompromised patients.

## Introduction

*Malassezia* is a genus of lipid-dependent yeast known to be a commensal on the skin of humans and other animals. However, these yeast species are primarily involved in skin infections such as pityriasis versicolor and seborrheic dermatitis [1]. In rare cases, they cause bloodstream infections, mainly in patients receiving lipid parenteral nutrition [2], and are involved in cases of infectious endocarditis [3,4].

**Data availability statement:** The data supporting the findings of this study are openly available in the Méditerranée Infection data repository at https://www.mediterranee-infection.com/acces-ressources/donnees-pour-articles/data-gut-hiv/; https://doi.org/10.35081/r4hx-g383.

**Funding:** ANR-10-IAHU-03 Région Provence Alpes Côte d'Azur, ERDF PRIMI the funders had no role in study design, data collection and analysis, decision to publish, or preparation of the manuscript.

**Competing interests:** The authors have declared that no competing interests exist.

Recent advances in sequencing technologies have provided new insights into the different niches of *Malassezia* species. It appears that the distribution of these yeasts is not restricted to the skin, but is ubiquitous in the human body [5]. In many studies on gut mycobiota based on deoxyribonucleic acid (DNA) sequencing, *Malassezia* has been found to be one of the most abundant and prevalent fungi in this biotope [6–11].The presence of *Malassezia* yeasts in the digestive tract raises many questions regarding their physiological and fastidious nature. Some authors have suspected that the *Malassezia* DNA detected in the digestive tract is contaminated [7].This hypothesis seems to have been dismissed as the findings have been reproduced in many studies conducted in various laboratories worldwide. A metagenomic study based on the internal transcribed spacer 2 (ITS2) barcode in a large cohort of healthy volunteers detected *M. restricta* in 88% of 317 samples, and its presence was stable in 78% of 129 healthy volunteers over the course of approximately one year [8].These authors concluded that *Malassezia* is a resident commensal and a part of the human core gut mycobiome. In the context of human diseases, *Malassezia* has been shown to be associated with Crohn's disease and exacerbated inflammation in a mouse model [12].It has also been shown to be significantly more abundant in the digestive tract of patients with colorectal cancer [13,14]and accelerated pancreatic oncogenesis [15].A metagenomic study reported a significantly higher prevalence of *M. restricta* in 31 persons living with HIV (PLWH) than in 12 healthy volunteers [10]. The authors advocate that further research should aim to understand whether *Malassezia* is an opportunistic pathogen in the digestive tract of HIV-infected patients. Interestingly, most PLWH in this study had a low CD4 T cell count of ~200 cells/mm3 [10].HIV infection triggers a massive depletion of CD4 T cells in the gastrointestinal tract, which is a major site of viral replication [16].

Another question concerns the viability of *Malassezia* yeasts detected in the gastrointestinal tract. The DNA-based methods used in the metagenomic studies cited above cannot distinguish between DNA originating from dead and living yeasts. Owing to their fastidious nature, *Malassezia* do not grow on traditional mycological media, and *Malassezia*-specific culture media are rarely used in digestive mycobiota studies. Furthermore, a negative stool culture can result from several factors, and this finding does not indicate that the microorganism is alive in the gastrointestinal tract. Using complementary culture methods is key to answering the question of viability. Several methods have been used to study cell viability based on membrane integrity. Among them, Propidium MonoAzide (PMA), a DNA-chelating compound that cannot enter the membranes of living cells, has been widely used in a variety of biological samples, including stools [17–19].PMA is usually coupled with real-time polymerase chain reaction (PCR) or high-throughput sequencing to quantify the proportion of viable microorganisms within a sample [20].To the best of our knowledge, no study has used multiple approaches to determine the viability of *Malassezia* in the gastrointestinal tract. This study aimed to characterize and assess the influence of HIV infection and HIV-associated immunodeficiency on human gut *Malassezia* yeast communities and to assess whether the *Malassezia* DNA detected stems from living yeasts.

## Patients and methods

### Participant recruitment and sample collection

Patients in this study were informed of the study protocol and were included in the infectious diseases outpatient ward of the Institut Hospitalo-Universitaire Méditerranée Infection (IHU-MI) at the La Timone University Hospital in Marseille, France. Between 08/11/2019 and 08/9/2021, 33 volunteers were included in the study, including 23 PLWH and ten healthy (not HIV-infected) controls. The inclusion criteria were as follows: age of at least 18 years and no history of systemic or local antifungal treatment in the last 30 days. In this longitudinal cohort study, five visits over a minimum period of one year were planned. At each visit, the participants had to provide a stool sample less than 24 hours old, collected in a sterile pot either at home or on site. The stool samples were processed for culture and viability PCR within an hour of receipt in the laboratory. Occasionally, a dietary questionnaire was completed in the previous seven days as well as information on the use of antibacterial or antifungal treatments.

### Cultures and identification

Liquid stool samples were plated directly, and ~4 g of solid stool sample was diluted in 4 mL of sterile saline. 100 μl of the samples were plated onto BBL CHROMagar Candida Medium plates (Becton Dickinson GmbH, Heidelberg, Germany), four FastFung medium plates [21,22], and four modified Dixon medium plates [23].The FastFung and modified Dixon media plates were each supplemented with 0.5 g/L cycloheximide (CliniSciences, Nanterre, France) and 4 mg/L rapamycin (CliniSciences, Nanterre, France) to control the growth of undesirable fungi, as previously reported [24], and with 0.5 g/L chloramphenicol (Sigma-Aldrich, Saint-Quentin Fallavier, France) and 0.03 g/L colistin (Sigma-Aldrich) to inhibit bacterial growth. All plates were placed in plastic bags, incubated aerobically at 30 °C from three to 15 days, and examined daily to assess the growth of fungal colonies. The colonies were identified using MALDI-TOF MS on a Microflex™ machine (Bruker Daltonics GmbH &Co, Bremen, Germany) with MALDI Biotyper™ software (Bruker Daltonics), as previously described [25,26].Between one and three colonies with similar aspects were identified on each plate. If the LogScore identification score was below 1.9, DNA sequencing-based identification was performed as previously described [25].

### PMA treatment and real-time-PCR

PMA (CliniSciences, Nanterre, France) aliquots of 1 mg were dissolved in 98 μl DNA and nuclease-free water (QIAGEN, Les Ullis, France) to achieve a 20 mM concentration. This stock solution was either used directly or stored at -20 °C in the dark. Stool samples were homogenized in 5 mL of sterile saline solution at 0.2 g of stool. For PMA treatment, 1.25 μL of PMA stock solution was added to Eppendorf Tubes Clear-Lock ClearLine™ (Dominique Dutscher, Bernolsheim, France) containing 500 μL of the sample, to achieve a 50 μM final PMA concentration, following the manufacturer's recommendations (Biotium Inc., USA). An equivalent volume of sterile saline was added to the untreated samples. PMA photoinduced cross-linking was initiated by ten minutes of incubation in the dark with occasional mixing, and the samples were then exposed to LED light (URSING Lampe, 12000LM 3 × XML T6 LED) at a distance of 15–20 cm for 20 min. After exposure to light, the samples were pelleted by centrifugation at 13 000 rpm for 10 min. The pellets were suspended in 350 μL of G2 lysis buffer with the addition of glass powder (Sigma-Aldrich, ref. G4649-500g) and mechanical lysis was performed with a FastPrep-24™5G V. 6005.1 (MP Biomedicals, LLC Santa Ana CA, USA) at six m/s for 40 s, followed by incubation at 100 °C for ten minutes. After centrifugation at 10 000 g for one minute, 200 μL of the supernatant was recovered and 10 μL of proteinase K was added, followed by two hours of incubation at 56 °C. DNA was extracted using the EZ1 DNA Tissue Kit (Qiagen GmbH, Hilden, Germany), according to the manufacturer's instructions. DNA was eluted in a total volume of 100 μL, 10 μL of DNA was used for PCR, and the rest was stored at -20 °C for further ITS and 16S metabarcoding analysis.

Real-time PCRs was performed using *Malassezia* 5.8S/ITS2 rRNA gene-specific primers and probes, as previously described [27,28]. The PCR reactions consisted of 10 µL of master mix (Roche Diagnostics GmbH, Mannheim, Germany), 0.5 µL of each primer, 0.5 µL of probe, 3.5 µL of distilled water and 5 µL of DNA in a total volume of 20 L. PCR conditions were ten minutes at 95 °C, 40 cycles of ten seconds at 95 °C and 30 seconds at 60 °C. The reactions were performed using a CFX96™ real-time PCR detection system (Bio-Rad, Life Science, Marnes-la-Coquette, France). Standard curves were constructed using serial dilutions of the plasmids (Eurogentec, Seraing, Belgium) corresponding to the amplified region. A mixture of DNA-free amplification reactions was used as a negative control. All reactions were performed in duplicate, and a sample was considered positive if the cycle threshold (Ct) value was below 38.

## ITS and 16S metabarcoding

The ITS1 and ITS2 regions were each amplified in triplicate, at 52 °C or 55 °C hybridization temperatures, following a previously described procedure [9].The 25 µl amplification reaction mix consisted of 12.5 µl AmpliTaq Gold master mix, 0.75 µl of each primer (Eurogentec, Seraing, Belgium), 6 µl distilled water, and 5 µl DNA template. The PCR cycling conditions were as follows: 95 °C for ten minutes, 40 cycles of 95 °C for 30 s, (55 °C or 52 °C) for 30 s, and 72 °C for one minute, followed by a five minute final extension step at 72 °C. For each sample, the amplicons of each ITS1 and ITS2 PCR replicate at the two temperatures were pooled. For bacteria, the 16S "V3-V4" regions were amplified using the primers with overhang adapters described herein [29].The amplification reaction mix consisted of 12.5 µl de Kapa HiFi HotStart ReadyMix 2x (Kapa Biosystems Inc., Wilmington, MA, USA), 0.5 µl of each primer (Eurogentec, Seraing, Belgium), 1.5 µl of distilled water, and 10 µl of DNA template for 25 µl volume. PCR cycling conditions were as follows:95 °C for three minutes, 45 cycles of 95 °C for 30 s, 55 °C for 30 s, and 72 °C for 30 s, followed by a five minute final extension step at 72 °C. PCR products were then purified using AMPure beads (Beckman Coulter Inc., Fullerton, CA, USA), quantified using high sensitivity Qubit technology (Beckman Coulter Inc., Fullerton, CA, USA), pooled, and barcoded before sequencing for ITS and 16S rRNA on a MiSeq system (Illumina, Inc., San Diego CA 92121, USA) with a paired-end strategy, as previously described [29,30].

## Bioinformatics and statistical analysis

Bioinformatic analyses of the ITS reads were performed using the PIPITS automated pipeline, which was developed for the analysis of ITS sequences from Illumina sequencing [31]. The OTU table was completed by manually analysing the sequences obtained from the operational taxonomic units (OTU) from the ITS1 and ITS2 regions via the standard Basic Local Alignment nucleotide Search Tool (BLASTN) search, as previously described [30].The criteria for taxonomic assignment of fungal OTUs were established as follows: percentage of identity (PID) >97% assignment to the species level; PID between 95% and 97%: assignment to genus; PID between 90% and 95%: assignment to the family; PID<90%: assignment to the kingdom. Sequences that were well assigned taxonomically to the family or genus level and not to the species level, for example, were noted as "unclassified" at the species level. For 16S sequences, the reads were analysed using the MetaGX and VSEARCH tools, as previously described [29].The SILVA database and the Culturomics and Diagnostics in-house databases were used for the taxonomic assignment of bacterial OTUs. The criteria for taxonomic assignment of bacterial OTUs were established as follows: 1) presence of one or more BLAST hits associated with a reference sequence (100% coverage; identity >97% corresponds to the assignment of OTUs to the species associated with the best BLAST hit); 2) presence of less relevant BLAST hits (identity between 95% and 97%: assignment to the genus level; between 90% and 95%: assignment to the family level;<90%: assignment to the kingdom level) with the creation of a putative species in each case; and 3) no BLAST hits (creation of putative new bacterial species).

Alpha diversity, the number of observed OTUs, and Shannon diversity were computed using R version 4.2.1. A comparison of alpha diversity between PLWH and healthy controls was performed using the Wilcoxon test. Beta diversity, measured by Bray-Curtis dissimilarity, was calculated using the vegdist function in the R package *vegan*. A Permutational

Multivariate Analysis of Variance (PERMANOVA) test was performed using the anosim function in the R package *vegan* to determine whether HIV status and immunosuppression explained the composition of the community. The relative abundances of bacterial and fungal taxa between the two groups were assessed by combining OTUs designating the same phylum, genus, or species. The genera *Aspergillus* and *Penicillium* were grouped into sections because of the lack of clear identification using ITS. Statistical significance was set at *p* value <0.05.

## Ethical approval

The study protocol was conducted in accordance with the Helsinki Declaration and received ethical clearance from the "Comité de Protection des Personnes Ile de France II" ethics committee (No. 19.05.29.69947 RIPH3, dated October 21, 2019). Participants provided written informed consent.

## Results

### Participant characteristics

The demographic and clinical characteristics of the PLWH and healthy volunteers upon inclusion are presented in Table 1. Of the 23 patients with HIV, ten patients were immunocompromised and 13 were non-immunocompromised. None of the patients reported receiving antifungal or antibiotic therapies. However, all PLWH are receiving highly active antiretroviral therapy. None of the healthy volunteers reported any diseases. The COVID-19 pandemic disrupted the study workflow, and many PLWH were lost to follow-up. Notably, two immunocompromised (IC) PLWH died. We grouped the two PLWH groups to analyse the follow-up data. There were 23 patients living with HIV at Visit 1 and 13 at Visit 2, including four IC patients, and eight from Visits 3–5, including two IC patients. All ten healthy volunteers participated in all visits.

### Fungal isolation

At Visit 1, the positive culture rates were 80%, 90%, and 77% in healthy volunteers, IC, and non-IV PLWH, respectively. Of the 22 fungal species, seven (*Pichia kudriavzevii*, *Aspergillus niger*, *Candida orthopsilosis*, *Rhodotorula mucilaginosa*, *Candida dubliniensis*, *Aspergillus tubingensis* and *Candida boidinii*) were isolated only from healthy volunteers and ten species from PLWH, of which five species (*Mucor velutinosus*, *Kluyveromyces lactis*, *Malassezia furfur*, *Scedosporium boydii* and *Candida lusitaniae*) were isolated from IC PLWH and five species (*Saccharomyces cerevisiae*, *Aspergillus flavus*, *Penicillium corylophilum*, *Chaetomium megalocarpum*, and *Penicillium commune*) were isolated from non-IC PLWH (Fig 1A). Notably, *M. furfur* was isolated from a 51-year-old patient with long-standing immunosuppression and a CD4 T cell count of 134. The mean colony forming units of *M. furfur* on FastFung and modified Dixon media were 18 and 25, respectively. Looking at the mycobiota

**Table 1. Participant characteristics upon inclusion.**

| Characteristic | Healthy | HIV + CD4 < 200 | HIV + CD4 > 500 |
|---|---|---|---|
| | (n = 10) | (n = 10) | (n = 13) |
| Age (years), mean ± SD | 34.6 ± 14.3 | 47.5 ± 8.6 | 54 ± 9.5 |
| Sex: Male, Female | 8, 2 | 8, 2 | 9, 4 |
| Antiretroviral Therapy – no (%) | 0 (0%) | 10 (100) | 13 (100) |
| Viral load (enrolment) | NA | | |
| HIV RNA undetectable – no. (%) | NA | 5 (50) | 6 (46) |
| HIV RNA detectable – no. (%) | NA | 5 (50) | 7 (54) |
| CD4 count (cells/mm³), mean ± SD | **1050 ± 424[a]** | **105.9 ± 48.8** | **701.7 ± 228.6** |
| CD8 count (cells/mm³), mean ± SD | 555.3 ± 230.4[a] | 838 ± 462.4 | 863.5 ± 369.5 |

[a]From eight healthy subjects, NA: not applicable

**A- Visit 1**

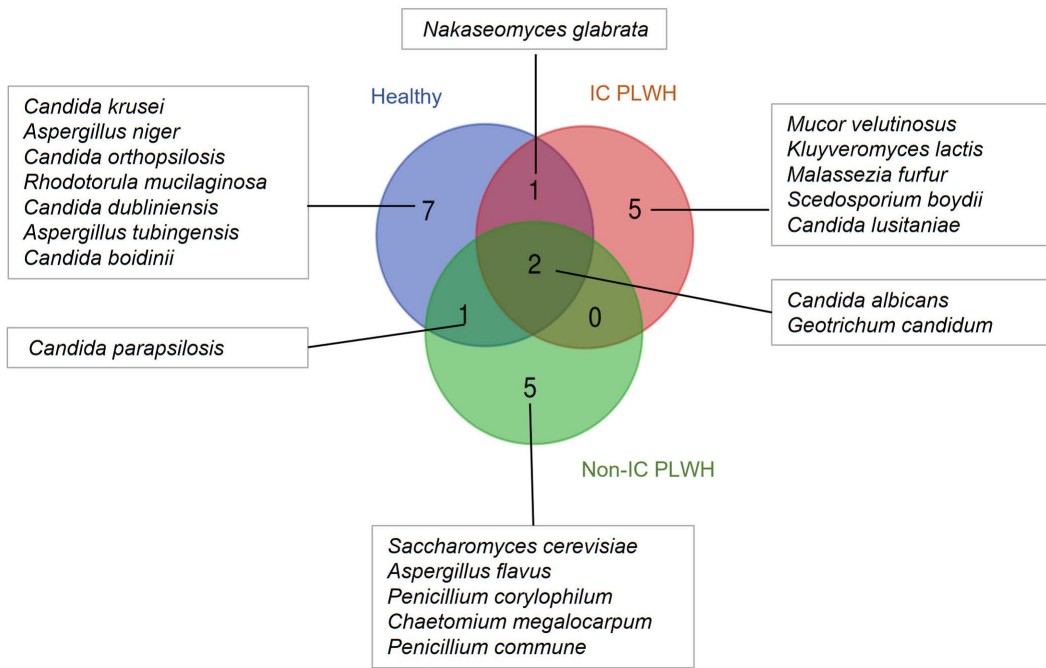

**B- All visits**

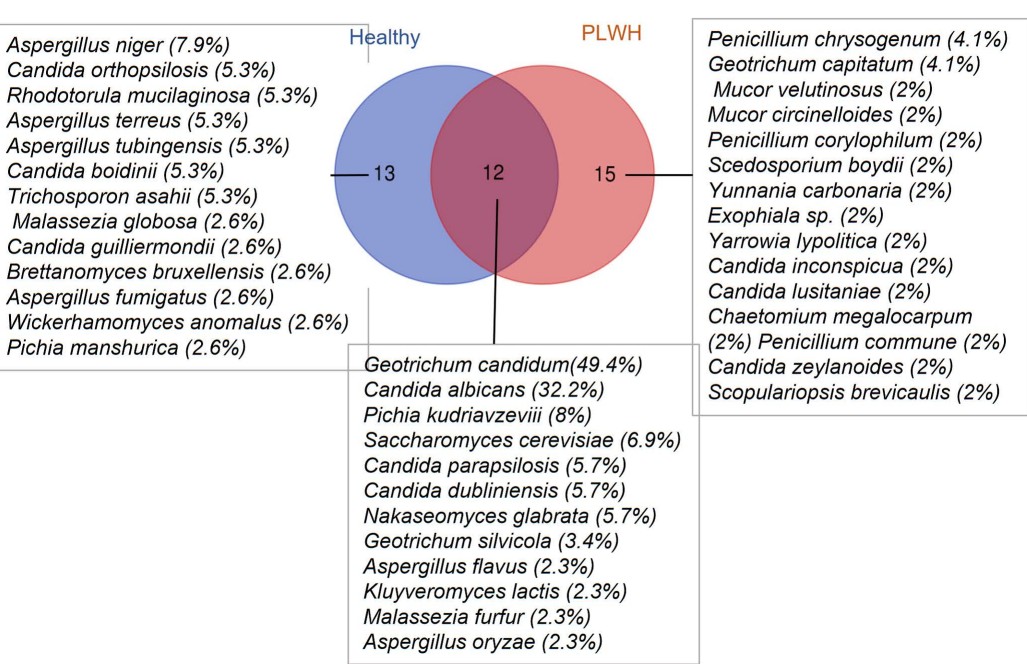

**Fig 1. Venn diagram of the diversity of fungal species isolated from immunocompromised (IC) or non-IC persons living with HIV (PLWH) and healthy volunteers at Visit 1 (A) and over all visits (B). All isolated strains were successfully identified by MALDI-TOF MS, with the exception of one _Chaetomium megalocarpum_ strain that was identified by DNA sequence analysis.**

shared by healthy volunteers only, *P. kudriavzevii* was isolated twice, whereas the other species were isolated once. In PLWH, *S. cerevisiae* was isolated three times, whereas the other species were isolated once. *C. albicans* and *G. candidum* were isolated from 37.5% vs. 25%, 55.6% vs. 22%, and 20% vs. 30% of healthy volunteers and IC and non-IC PLWH, respectively. Notably, *C. albicans* colonies could occasionally be detected between five and seven days, despite supplementing both Fast-Fung and modified Dixon media with rapamycin. CHROMAgar medium enabled the isolation of 86.4% (19/22) of the species, followed by FastFung and modified Dixon media, with 22.7% (5/22) and 9.1% (2/22), respectively.

We then analysed the available stool samples collected at the five follow-up visits to determine the stability of the gut microbiota. The positive culture rates were 76% (38/50) and 81.6% (49/60) in healthy volunteers and patients living with HIV, respectively. Most of the isolates (73.8%, 31/42) were cultured on CHROMAgar, followed by FastFung (38.1%, 16/42), and modified Dixon media (30.9%, 13/42). The fungal species isolated and their isolation frequencies are shown in Fig 1B. Analysis of the fungal species isolated only in one group of participants showed that *A. niger* (7.9%) was the most frequent species in healthy volunteers, whereas *P. chrysogenum* (4.1%) and *G. capitatum* (4.1%) were the most frequent in PLWH (Fig 1B). However, *Geotrichum candidum* (49.4%) and *C. albicans* (32.2%) were the most common fungal species shared by both groups. Some fungal species, including *C. albicans*, *G. candidum* and *C. dubliniensis*, were shown to be relatively stable within individuals during follow-up. *M. furfur* (one colony) and *M. globosa* (one colony) were isolated at Visit 4 from two healthy volunteers.

## Viability PCR

While the negative controls were always negative, three stool samples were PCR-positive, and all these samples originated from IC PLWH. In both patients (P1-200_Visit5 and P6-200_Visit1), PCR amplified only the untreated samples, and not the PMA-treated samples, suggesting that there were dead *Malassezia* yeasts (Fig S1A and 2B, supplementary data). However, *M. furfur* was isolated in culture from one of them (P6-200_Visit1). For the third patient (P2-200_Visit1), no difference was observed between the samples treated with PMA and the untreated samples, suggesting the presence of viable *Malassezia* yeasts (Fig S1C). However, the culture tested negative for *Malassezia*. The P2-200 patient was lost to follow-up while the P6-200 patient died; therefore, we were unable to obtain additional samples to assess stability.

## ITS and 16S metagenomic sequencing

From the 33 samples collected at Visit 1, ITS metagenomics yielded 506 151 reads and 573 single OTUs, including 232 451 reads and 304 single OTUs from ITS1 and 273 700 reads and 269 single OTUs from ITS2. The phylum Ascomycota was the most abundant (95.6% in ITS1 and 98.8% in ITS2) followed by Basidiomycota (4.1% in ITS1 and 1.0% in ITS2). Based on 16S rRNA gene sequencing, only 917 678 reads and 1125 OTUs were obtained after bioinformatic analyses. While 69.6% of the reads were unassigned, the most abundant phylum was Bacteroidetes (12.4%), followed by Firmicutes (11.1%) and Proteobacteria (2.8%). In ITS1, significant differences were found only in Shannon diversity between healthy controls and HIV-immunocompromised patients (Wilcox test; $P = .029$) as well as healthy controls and non-IC PLWH (Wilcox test; $P = .049$) (Fig 2B). However, the number of OTUs observed and Shannon diversity index did not significantly differ between PLWH and controls in both the ITS2 and 16S rRNA results (Fig 2). Gut fungal and bacterial community analyses revealed heterogeneous structures in healthy controls compared to non-IC PLWH with the ITS1 (ANOSIM test, $R = 0.09888$, $P = .0275$), ITS2 (ANOSIM test, $R = 0.09802$, $P = .0246$), and 16S (ANOSIM test, $R = 0.178$, $P = .0014$) barcodes (Fig 3). However, no clustering was observed in HIV-immunocompromised patients.

While the phylum Ascomycota dominated the gut fungal community, Bacteroidetes and Firmicutes dominated the bacterial community (Fig 4A and 4B). The genus *Candida* was the most abundant with the ITS1 barcode, whereas *Saccharomyces* was the most abundant with the ITS2 barcode in all samples (Fig 4). Surprisingly, the abundance of the genus *Candida* was strongly increased in IC PLWH in both ITS1 and ITS2 (Fig 4), suggesting an effect of immunity on *Candida* gut colonisation. *S. cerevisiae* and *C. albicans* were the most prevalent species, with frequencies of 93.9% and 87.9% in

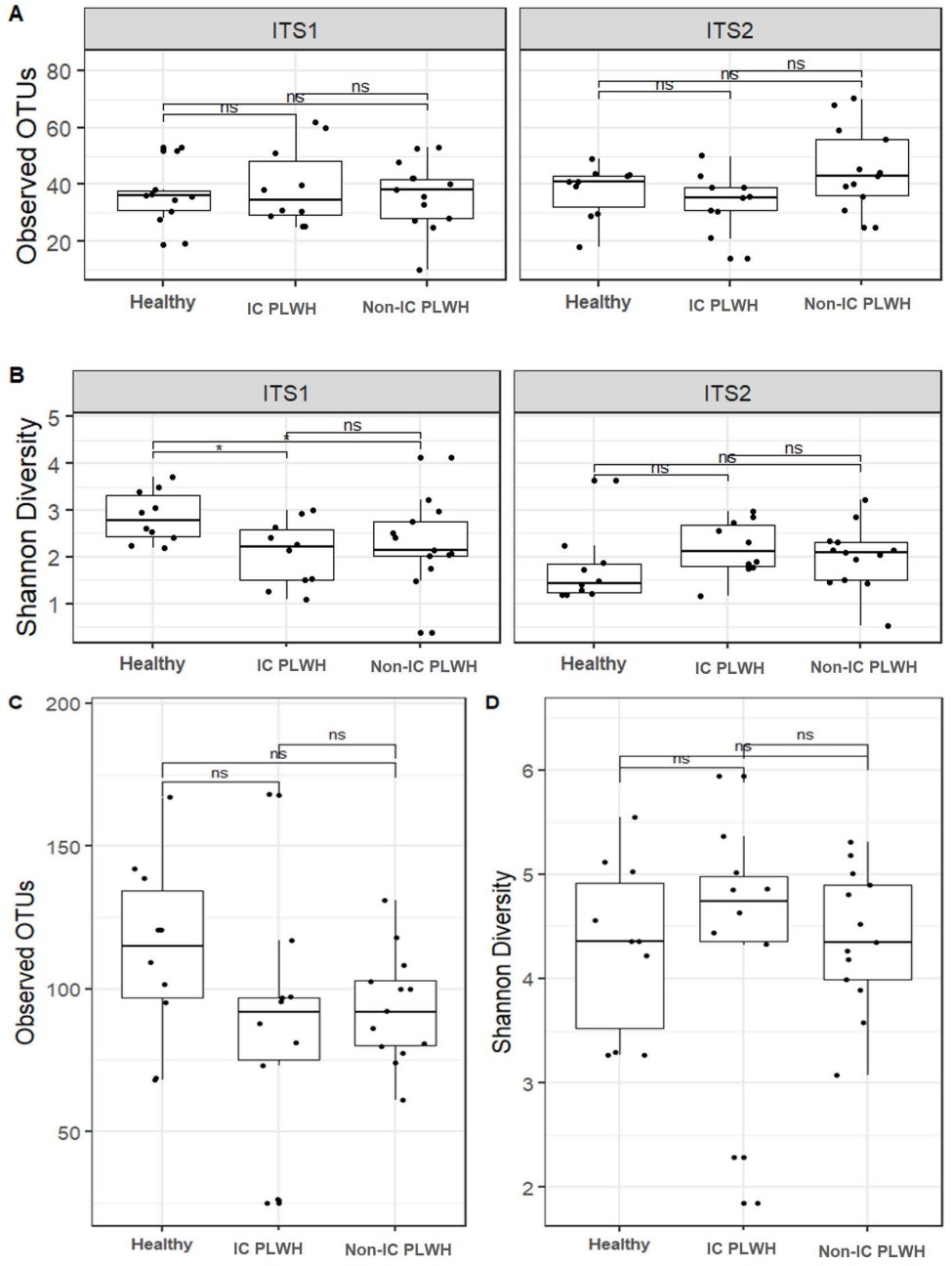

**Fig 2. Visit 1 fungal and bacterial alpha diversity.** Observed OTUs and Shannon diversity values in ITS (**A** and **B**) and 16S (**C** and **D**), respectively. The comparison was performed from samples of each volunteer. *$P < .05$; ns: not statistically significant ($P \geq .05$); Wilcoxon test. IC, immunocompromised; non-IC, non-immunocompromised; PLWH, persons living with HIV.

ITS1, respectively. In ITS2, *S. cerevisiae* was the most prevalent species (87.9%). Owing to a lack of sequence assignment in ITS2, some species were not well identified. For example, *Saccharomyces* sp. (100% of samples), Aspergillaceae (90.9%), *Candida* sp. (72.7%), and *Debaryomyces* sp. (63.6%) were more prevalent and abundant.

Additionally, by analysing the number of *Malassezia* sequences detected, we found that the abundance of *Malassezia* was strongly increased in IC PLWH compared to non-IC PLWH and healthy controls (Table 2), also suggesting an effect of

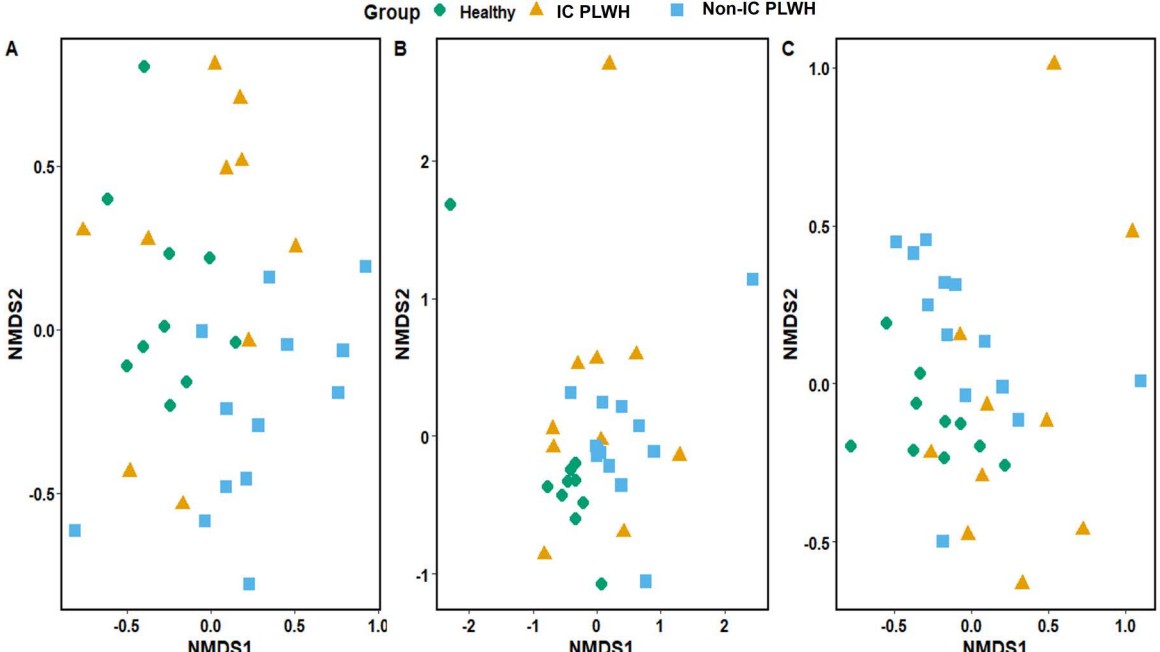

**Fig 3. Visit 1 fungal and bacterial communities.** Ordinations of community composition based on Bray-Curtis distance for fungal (**A** ITS1, and **B** ITS2) and bacterial (**C**) gut communities in patients living with HIV (PLWH) and healthy controls. The comparison was performed on the samples of each participant.

immunity on *Malassezia* gut colonization. Interestingly, IC PLWH with positive *Malassezia* PCR results had more *Malassezia* reads in metagenomics. At the species level, *M. restricta* followed by *M. globosa* were the most frequent species in all individuals, with a frequency of 72.7% vs. 24.2% in ITS1, and 60.6% vs. 33.3% in ITS2, respectively. For bacteria, the distribution of the relative abundance of the genera was relatively homogeneous in each group (Fig 4B).

An analysis of the variability of the fungal and bacterial gut microbiota was performed for two successive time samples by calculating the Bray-Curtis dissimilarity. A comparison was then performed between and within PLWH and between and within healthy controls. However, no significant difference was found between PLWH and healthy controls (Fig S2A and S2B).

To assess the stability of the gut mycobiome both in PLWH and in controls, we analysed 90 stool samples, including 40 samples from PLWH and 50 samples from controls, collected from five visits for each participant. We found that intra-individual alpha diversity was unstable over time (Fig S3). The ITS1 analysis showed that *S. cerevisiae* (92.2% of all samples) and *C. albicans* (82.2%) were stable in the gut (Table 3) and were present in at least three samples from each individual. *S. cerevisiae* (86.7% of all samples) was also stable in ITS2 (Table 3). Unidentified species *Saccharomyces* sp. (100% of all samples), Aspergillaceae (81.1%), *Candida* sp. (70%), and *Debaryomyces* sp. (90%) were also found to be stable in the gut (Table 3). The high frequency of these species indicates that they are commensal and form a part of the core gut mycobiome. Focusing on the *Malassezia* genus, we found 37.8% and 44.4% prevalence within all samples using the barcodes ITS1 and ITS2, respectively. The intra-individual dynamics of *Malassezia* reads (plotted in Fig S3) showed a very unstable and relatively infrequent *Malassezia* spp. occurrence in the gut.

## Discussion

Previous gut mycobiota studies have detected significant numbers of *Malassezia* reads in the context of health [7,8]or disease, including HIV infection [9,10]. However, no study has investigated in depth the viability of the detected *Malassezia*

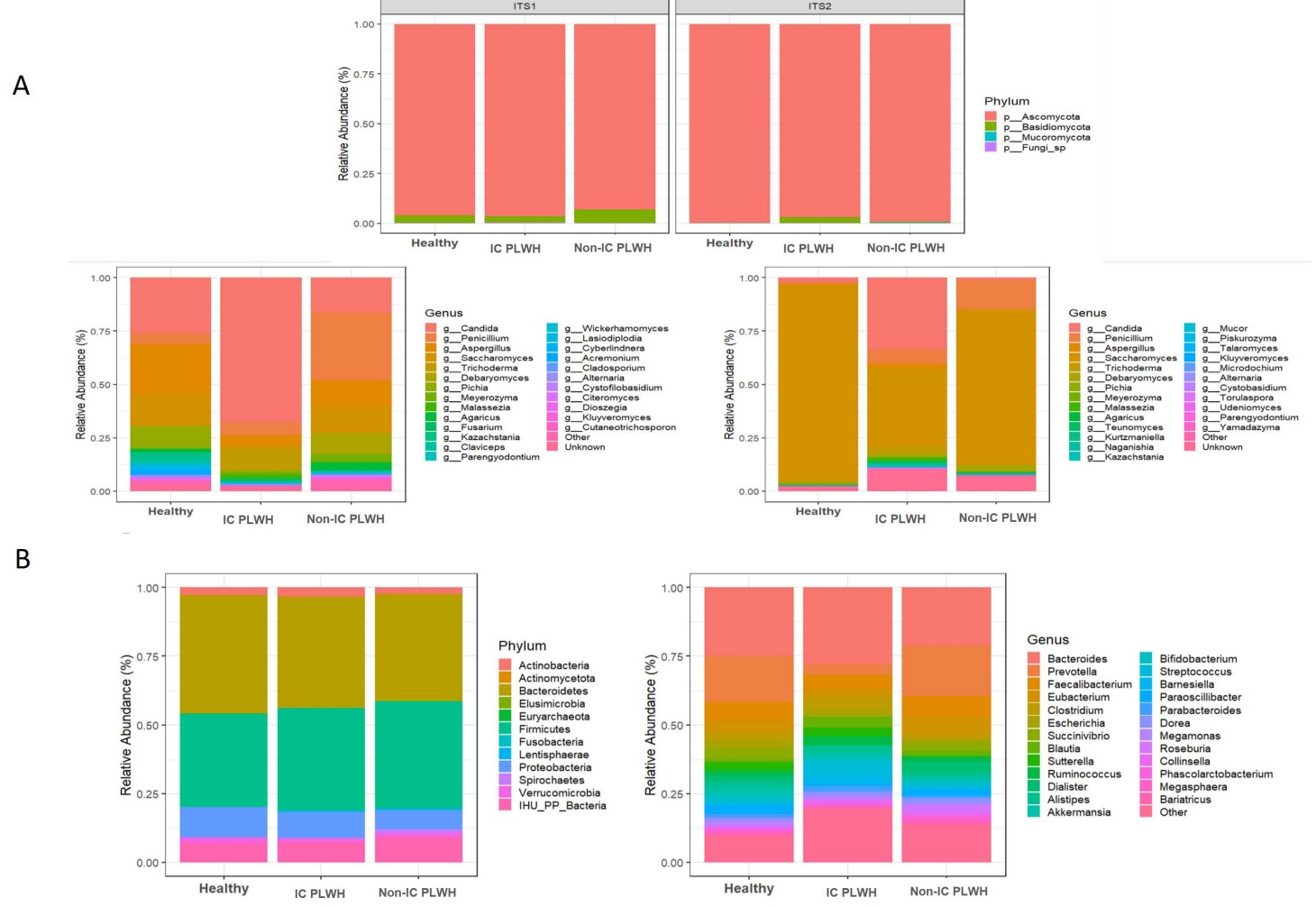

**Fig 4. Visit 1 relative abundance of fungi and bacteria at the phylum and genus level.** Relative abundance of phyla and the 25 most abundant fungal genera (**A**) (ITS1: left and ITS2: right) and bacterial genera (**B**). "Fungi sp." here represents unknown/unidentified fungal phylum. "Other" here represents the other genera. "Unknown" represents unidentified fungal genera. The abundance of unassigned 16S reads is not presented here. IC, immunocompromised; non-IC non immunocompromised; PLWH, persons living with HIV.

**Table 2. *Malassezia* reads detected in people living with HIV (PLWH) and healthy controls.**

| | *Malassezia* spp. reads | |
|---|---|---|
| **Study participants** | ITS1 (n = 2930) | ITS2 (n = 1347) |
| **Controls** | 97 (3.3%) | 90 (6.7%) |
| **PLWH** | | |
| **-Immunocompromised** (CD$_4$ < 200/μl) | 2654 (90.6%) | 1081 (80.2%) |
| **-Immunocompetent** (CD$_4$ > 500/μl) | 179 (6.1%) | 176 (13.1%) |

—

**Table 3. Top 20 most prevalent taxa using ITS1 or ITS2 metabarcoding characterization of the fungal gut community in 90 stool samples.** Table indicates the percentage of samples in which the species was identified.

| ITS1 | | ITS2 | |
|---|---|---|---|
| *Saccharomyces cerevisiae* | 92.2 | *Saccharomyces* sp. | 100 |
| *Candida albicans* | 82.2 | *Debaryomyces* sp. | 90 |
| *Debaryomyces prosopidis* | 82.2 | *Saccharomyces cerevisiae* | 86.7 |
| *Penicillium* Sect. Fasiculata | 76.7 | Aspergillaceae | 81.1 |
| *Saccharomyces* sp. | 71.1 | *Saccharomycetaceae* sp. | 75.6 |
| *Pichia kudriavzevii* | 57.8 | *Candida* sp. | 70 |
| *Candida africana* | 54.4 | *Penicillium* sp. | 60 |
| *Candida tropicalis* | 52.2 | *Candida sake* | 48.9 |
| *Aspergillus* Sect. Nigri | 48.9 | *Pichia* sp. | 42.2 |
| *Aspergillus* Sect. Fumigati | 47.8 | *Malassezia restricta* | 41.1 |
| *Penicillium* Sect. Chrysogena | 47.8 | *Aspergillus* Sect. Nigri | 37.8 |
| *Alternaria sp.* | 43.3 | *Cladosporium* sp. | 27.8 |
| *Cladosporium chasmanthicola* | 37.8 | *Alternaria sp.* | 21.1 |
| *Aspergillus* Sect. Flavi | 37.8 | *Aspergillus* Sect. Flavi | 20 |
| *Malassezia restricta* | 34.4 | *Torulaspora* sp. | 18.9 |
| *Penicillium* sp. | 28.9 | *Penicillium* Sect. Fasciculata | 18.9 |
| *Penicillium* Sect. Roquefortorum | 27.8 | *Hanseniaspora uvarum* | 16.7 |
| *Meyerozyma guilliermondii* | 27.8 | *Kluyveromyces lactis* | 16.7 |
| *Candida parapsilosis* | 25.6 | *Candida parapsilosis* | 15.6 |
| *Penicillium crustosum* | 25.6 | *Aureobasidium lini* | 14.4 |

yeasts and whether they colonize the digestive tract of IC PLWH in a stable manner. Using a *Malassezia*-specific culture medium, we first isolated many *M. furfur* strains from one IC PLWH's stool. At Visit 4, we isolated one colony each of *M. furfur* and *M. globosa* from the stools of two distinct healthy controls. Data on the isolation of *Malassezia* spp. from the gut are scarce in literature. To date, three studies have reported the isolation of *M. globosa*, *M. restricta*, and *M. pachydermatis* from single stool samples of four distinct individuals [23,32,33].One strength of our study was the use of *Malassezia*-specific culture medium. Our findings suggest that cultivable *Malassezia* within the human gut is rare. When analysing the diversity of non-*Malassezia* fungal species, we only isolated 13 taxa from healthy volunteers and 15 taxa from PLWH (Fig 1B). However, these taxa were isolated at a low frequency, and none were significantly associated with any of the groups of study participants. All participants shared a core mycobiome, with a dominance of *G. candidum* (49.4%) and *C. albicans* (32.2%). *G. candidum* is infrequently isolated in culture compared to *C. albicans* which is a human gut commensal and is frequently cultured [34].Most of the fungal species that we cultured have been reported to be isolated from human stool samples [9].

Using PMA-viability PCR, we amplified *Malassezia* DNA from three stool samples collected from IC PLWH. In two of them, PMA-viability PCR indicated the presence of non-living *Malassezia*, even though *Malassezia furfur* was cultured from only one of these samples. In the third patient, although the culture was negative, PMA-viability PCR indicated the presence of living *Malassezia*. The discrepancies between the PMA-viability PCR and culture results may be explained by the difficulties of *Malassezia* culture and the fact that distinct subsamples of the stools were used for PCR or culture. PMA-PCR and culture methods have distinct analytical sensitivities. Additionally, culturable *Malassezia* spp. were present in relatively low abundance. Thus, we can hypothesize that their distribution within the sample was highly heterogeneous, and the probability of detecting *Malassezia* spp. in all sub-samples was low. To our knowledge, this study is the first to use

PMA-viability PCR to assess the viability of *Malassezia* spp. detected in anatomical niches such as the digestive tract. PMA-viability PCR was used for various sample types [18,35–38].It has been used mainly in stool samples for the study of bacteria and cryptosporidia [18,19].

Only the Shannon diversity index with the ITS1 barcode differed significantly between PLWH and controls (Fig 2). The gut fungal community structures of non-immunosuppressed patients living with HIV and the controls were heterogeneous (Fig 3). In particular, the *Candida* genus was highly abundant in IC PLWH (Fig 3). Consistent with the ITS metabarcoding results, abundant colonies of *C. albicans* were cultured in 50% of our HIV-immunocompromised patients. A high abundance of *Candida* spp. in the gut of patients living with HIV has also been reported by Hamad et al. [10].However, the causes and clinical impact of this higher *Candida* spp. abundance in the gut of PLWH remain unknown. In line with Nash et al. [8], we found that *S. cerevisiae* and *C. albicans* were relatively stable in the gut, which suggests that these species belong to the human gut core mycobiome. While Nash et al. [8]reported similar findings regarding *M. restricta*, we observed a low prevalence of the *Malassezia* genus and erratic intra-individual dynamics of the abundance of *Malassezia* reads, which both indicate that this genus does not belong to the human core mycobiome.

In this study, we found a high abundance of *Malassezia* spp. reads in the gut of PLWH (Table 2), and we isolated abundant *M. furfur* colonies from one of the IC PLWH who also tested positive *Malassezia* by PCR. Our results suggest that further investigations are needed to understand the composition of the gut fungal community in PLWH. Notably, abundant *Malassezia* colonies were isolated from stool samples of one IC PLWH. Unfortunately, we had no follow-up samples because the patient died. Remarkably, our viability PCR results were positive only for IC PLWH. Untreated HIV infection is known to cause massive depletion of $CD_4$ T cells, and the digestive tract is one of the major sites of viral replication [16]. Therefore, our results warrant further investigation to better understand the potential pathogenic role of *Malassezia* in PLWH. Indeed, there is little clinical data on the effect of *Malassezia* on the digestive tract, but a pathogenic role of *Malassezia* in Crohn's disease [39]and pancreatic cancer [15]has been shown in mice.

Our study had several strengths. It analysed the dynamics of the gut *Malassezia* yeast community over time, something which has seldom been done, and metagenomic analysis provides insights into both fungal and bacterial gut communities associated with HIV infection. The main limitations of our study were the heterogeneity of the sample collection period, the follow-up visits of our participants, and the number of PLWH who were lost to follow-up. These minor deviations from the protocol were a consequence of the generalized disorganization of patient care during the COVID-19 pandemic. Another limitation was the relatively small sample size of our cohort owing to recruitment difficulties. Finally, many fungal and bacterial sequences were not well identified owing to the incompleteness of the fungal nucleotide databases currently available.

In conclusion, both our culture and PCR findings indicate that live *Malassezia* yeasts are infrequent in the human digestive tract and that HIV-related immunosuppression is likely to promote the presence of *Malassezia* spp. in the human gut. Further research is needed to clarify the role of *Malassezia* yeasts in the human gastrointestinal tract.

## Supporting information

**Fig S1. Viability PCR performed in three stool samples of HIV-immunocompromised patients.** Blue and red curves represent untreated (PMA-) and treated (PMA+) stools, respectively. In panels **A** and **B**, the PCR amplified only the untreated samples and not the PMA-treated samples, indicating the presence of dead *Malassezia* yeasts. In panel **C**, both PMA-treated and untreated samples were similarly amplified, indicating the presence of living *Malassezia* yeasts in the sample.
(TIF)

**Fig S2. Variability of the fungal and bacterial gut microbiota.** Comparisons of Bray-Curtis dissimilarity values between Visit 1 and Visit 2 samples donated by different participants (A) (between controls, n = 10) or different patients living with

HIV (PLWH B) (between PLWH, n = 13) and between samples donated by the same control (within healthy) or the same PLWH (within PLWH) for 16S and ITS. Bray-Curtis dissimilarity values range from 0 to 1, with 0 being the least dissimilar and 1 being the most dissimilar. ns: not statistically significant Wilcoxon test (p > 0.05).
(TIF)

**Fig S3. Dynamics of the fungal alpha diversity over time in stools samples of PLWH (HIV + , P) and healthy controls (H).** Detail of the observed OTUs (A, B) and Shannon diversity indices (C, D), by ITS1 (A, C) or ITS2 (B, D) metabarcoding, respectively.
(TIF)

## Author contributions

**Conceptualization:** Abdourahim Abdillah, Stephane Ranque.

**Formal analysis:** Stephane Ranque.

**Investigation:** Abdourahim Abdillah, Isabelle Ravaux, Saadia Mokhtari, Stephane Ranque.

**Methodology:** Abdourahim Abdillah, Isabelle Ravaux, Saadia Mokhtari, Stephane Ranque.

**Project administration:** Stephane Ranque.

**Resources:** Stephane Ranque.

**Supervision:** Stephane Ranque.

**Visualization:** Abdourahim Abdillah.

**Writing – original draft:** Abdourahim Abdillah.

**Writing – review & editing:** Isabelle Ravaux, Saadia Mokhtari, Stephane Ranque.

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
