## [Decision Letter · Decision Letter 0]

10 Oct 2024

PONE-D-24-36362Do Malassezia yeasts colonize the guts of people living with HIV?PLOS ONE

Dear Dr. Ranque,

Thank you for submitting your manuscript to PLOS ONE. After careful consideration, we feel that it has merit but does not fully meet PLOS ONE’s publication criteria as it currently stands. Therefore, we invite you to submit a revised version of the manuscript that addresses the points raised during the review process.

We look forward to receiving your revised manuscript.

Kind regards,

Kin Ming Tsui

Academic Editor

PLOS ONE

Journal Requirements:

1. When submitting your revision, we need you to address these additional requirements. Please ensure that your manuscript meets PLOS ONE's style requirements, including those for file naming. The PLOS ONE style templates can be found at https://journals.plos.org/plosone/s/file?id=wjVg/PLOSOne_formatting_sample_main_body.pdf and https://journals.plos.org/plosone/s/file?id=ba62/PLOSOne_formatting_sample_title_authors_affiliations.pdf 2. We note that the grant information you provided in the ‘Funding Information’ and ‘Financial Disclosure’ sections do not match.  When you resubmit, please ensure that you provide the correct grant numbers for the awards you received for your study in the ‘Funding Information’ section. 3. Thank you for stating the following financial disclosure: "ANR-10-IAHU-03 Région Provence Alpes Côte d’Azur, ERDF PRIMI"  Please state what role the funders took in the study.  If the funders had no role, please state: "The funders had no role in study design, data collection and analysis, decision to publish, or preparation of the manuscript." If this statement is not correct you must amend it as needed. Please include this amended Role of Funder statement in your cover letter; we will change the online submission form on your behalf. 4. Please note that in order to use the direct billing option the corresponding author must be affiliated with the chosen institute. Please either amend your manuscript to change the affiliation or corresponding author, or email us at plosone@plos.org with a request to remove this option. 5. Your ethics statement should only appear in the Methods section of your manuscript. If your ethics statement is written in any section besides the Methods, please move it to the Methods section and delete it from any other section. Please ensure that your ethics statement is included in your manuscript, as the ethics statement entered into the online submission form will not be published alongside your manuscript. 6. We notice that your supplementary figures are included in the manuscript file. Please remove them and upload them with the file type 'Supporting Information'. Please ensure that each Supporting Information file has a legend listed in the manuscript after the references list.

Additional Editor Comments:

This is an interesting manuscript that addresses the niche of Malassezia in gut. However, it is not written in a way that depicts its significance from the referees' report.

Please provide additional clinical data if possible on these HIV patients and to improve the logic and writings.

Reviewers' comments:

Reviewer's Responses to Questions

**Comments to the Author**

1. Is the manuscript technically sound, and do the data support the conclusions?

Reviewer #1: Yes

Reviewer #2: Partly

2. Has the statistical analysis been performed appropriately and rigorously? 

Reviewer #1: Yes

Reviewer #2: N/A

3. Have the authors made all data underlying the findings in their manuscript fully available?

Reviewer #1: Yes

Reviewer #2: No

4. Is the manuscript presented in an intelligible fashion and written in standard English?

Reviewer #1: Yes

Reviewer #2: No

5. Review Comments to the Author

Reviewer #1: Thanks for invitation. The authors showed a correlation between Malassezia and HIV. The manuscript was written well.

But I have some concerns.

1. In this study, were the main covariates for group comparisons limited to HIV status and absolute CD4 count, or were other common factors that could affect microbiota outcomes, such as age and diet, also adjusted for?

2. The authors analyzed both bacterial and fungal communities. Could the results be interpreted together rather than separately? I believe that, regardless of causality, there would be interactive effects between the two.

3. As the authors mentioned in the limitations, this is merely a descriptive study without any associated clinical outcomes, let alone the references to conditions like Crohn's disease (CD) mentioned in the paper. This is the study's biggest limitation.

4. As a clinician, I find that such descriptions offer very limited help for clinical practice. I am not very familiar with the French healthcare system. Is it possible to trace the gastrointestinal conditions of these patients, such as unformed stools or other related issues?

Reviewer #2: The manuscript titled “Do Malassezia yeasts colonize the guts of people living with HIV?” from Stéphane Ranque’s laboratory presents analyses of stool samples collected from a longitudinal cohort. The study aims to compare the effects of HIV infection on Malassezia in human gut using conventional culture methods, metagenomics and PMA-qPCR. While the data presented are limited due to the small sample size, they are intriguing and could serve as a foundation for further research on this topic. However, I recommend the authors significantly improve the way they present the data before publication.

Major points: The primary issue lies in the presentation of the data, which is not easy to follow. Here, I will highlight some points that the authors may consider for improvement:

1. Title: While I like the title, I think that the manuscript did not fully address the question it poses. Since the data is not strong enough to provide a definitive answer, I suggest slightly modifying the title to better align with the study's findings.

2. Abstract: The authors may need to reorganize the abstract. It should be concise, with fewer methodological details and a stronger focus on the findings and conclusions. Readers will likely expect a more engaging abstract based on the manuscript’s title. As it stands, I lost motivation because the abstract does not clearly address the question posed in the title.

3. Main results: This section faces similar issues as the abstract. For each subsection, it would be more effective to present the key findings in the subsection titles rather than describing the results as if outlining research steps. Readers are more interested in what the data tells, not just what the data is or what the authors did. Due to this presentation style, the primary focus on Malassezia is not clearly emphasized.

4. Figure 1: Author could include a cartoon or schematic to illustrate your sample collection and patient information. This will give the audience an overview of the patient samples, including how many were used for culture, sequencing, how many patients completed the full study, and how many dropped out.

In Figure 1A, it would be helpful to include the number of positive cultures for each species. Additionally, labeling "Visit 1" and "All visits" more clearly in Figures 1A and 1B would improve clarity.

The font styles and sizes are inconsistent across the figures. Please use a consistent Sans-serif font, and increase the font size to improve visibility.

There are many following figures based on only “Visit 1”. Authors can consider collecting them together in a big figure or indicate “Visit 1 analysis…” directly on these figures that would be easier to understand.

5. Figure S1: author may replace this figure with a table or bar plots showing the Ct values. The current figure is difficult to interpret due to the colors and lines, and it doesn’t provide much information. A table or bar plots would make it easier for readers to compare the data. The figure primarily needs to indicate which samples are PCR-positive (+) and which are negative (-). The current presentation is unnecessarily complex and takes too much time for readers to grasp the simple results.

6. Figure 2: Although Figure 2B shows two significant comparisons, the Shannon diversity analysis may be biased due to the small sample size. Therefore, the figure essentially indicates no significant difference between the samples. The title of the figure should reflect this main result. Additionally, "Visit 1" should be indicated earlier in the title or directly on the graph for better clarity and understanding.

7. Figure 4: The font size is too small and needs to be increased for better readability. Additionally, the organization of this figure could be improved. I suggest arranging the x-axis based on health conditions, from healthy to severe disease. This would make it easier to compare how the microbiome changes across different health conditions.

8. Table 2: The table lacks meaningful impact unless the authors consider how many samples are positive and the variability of read counts between samples.

9. Overall, there are so many descriptions, statistical number (percentage) that describe in the main text, however, lack of conclusion from each section and even each paragraph make the manuscript very hard to follow. I could not really catch the main finding of each section. Notably, the focus in this manuscript should be answering “Do Malassezia colonies the guts of HIV patients?” which are blurry because of many details and descriptions about statistical numbers of microbiome.

Therefore, while the data is valuable, the authors need to better organize the manuscript and focus on presenting the findings rather than just describing the data. Detailed descriptions should primarily be presented in the Figures and Tables, while the Results section should briefly summarize, analyze, and provide clear conclusions. The Discussion section is in good shape for now.

Minor points:

1. Citations should be added before the sentence-ending period (".").

2. Line 112: Please specify the volume used for plating on each plate.

3. Will the metagenomic sequencing data be made publicly available? If there are no restrictions from the ethical agreement, the authors should consider depositing the data in a public repository.

4. Line 210: p value < 0.05

5. Line 227: Wee

6. Line 257: This is the first appearance of the genus “G.”.

7. Line 345: Aspergillaceae is a family, not a genus.

8. The authors should abbreviate terms such as "HIV-immunocompromised patients," "non-immunocompromised," "non-immunocompromised patients living with HIV," "immunocompromised and non-immunocompromised patients living with HIV," "CD4 < 200," and "CD4 > 500" for consistency and easier comparison of data. These abbreviations should be used consistently throughout the manuscript and figures to make the content easier to follow for a broad readership.

9. Table 3: please check alignment of some lines.

6. PLOS authors have the option to publish the peer review history of their article (what does this mean? ). If published, this will include your full peer review and any attached files.

**Do you want your identity to be public for this peer review?** For information about this choice, including consent withdrawal, please see our Privacy Policy .

Reviewer #1: No

Reviewer #2: No

---

## [Author Response · Author response to Decision Letter 1]

7 Mar 2025

Journal Requirements:

Response: Done.

3. Thank you for stating the following financial disclosure: "ANR-10-IAHU-03 Région Provence Alpes Côte d’Azur, ERDF PRIMI"

Response: This has been added in the financial disclosure section.

Response: Included in the cover letter.

Response: corresponding author email OK

Response: OK

6. We notice that your supplementary figures are included in the manuscript file. Please remove them and upload them with the file type 'Supporting Information'. Please ensure that each Supporting Information file has a legend listed in the manuscript after the references list.

Response: Done

Additional Editor Comments:

This is an interesting manuscript that addresses the niche of Malassezia in gut. However, it is not written in a way that depicts its significance from the referees' report.

Please provide additional clinical data if possible on these HIV patients and to improve the logic and writings.

Response: Thank you for this interesting suggestion. However, our study design was based on the comparison of 'healthy' controls and people living with HIV, among whom we distinguished between immunocompromised and non-immunocompromised on the basis of CD4 cell count. The characteristics of these participants are detailed in Table 1. We have described in the manuscript the clinical condition of the patient from whose stool we isolated numerous Malassezia furfur colonies. Otherwise, the results were too heterogeneous to analyse all patient characteristics.

Comments to the Author

Reviewer #1: Thanks for invitation. The authors showed a correlation between Malassezia and HIV. The manuscript was written well.

But I have some concerns.

1. In this study, were the main covariates for group comparisons limited to HIV status and absolute CD4 count, or were other common factors that could affect microbiota outcomes, such as age and diet, also adjusted for?

Response: We agree that many factors, including age and diet, affect the microbiome. However, our sample size was relatively small and we lacked the statistical power to account for the effect of multiple covariates.

2. The authors analyzed both bacterial and fungal communities. Could the results be interpreted together rather than separately? I believe that, regardless of causality, there would be interactive effects between the two.

Response: This suggestion runs counter to the reviewer’s recommendations for simplifying the manuscript.

3. As the authors mentioned in the limitations, this is merely a descriptive study without any associated clinical outcomes, let alone the references to conditions like Crohn's disease (CD) mentioned in the paper. This is the study's biggest limitation.

Response: We agree that our study is a descriptive study and by design we did not investigate clinical outcomes. However, in the Introduction and Discussion sections, we mentioned data from experimental studies, mainly in mice, suggesting a potential pathogenic effect of Malassezia in Crohn's disease to rationalise our approach, which included isolation of Malassezia from stool samples. This in particular was a real mycological challenge as we had to develop a specific culture medium.

4. As a clinician, I find that such descriptions offer very limited help for clinical practice. I am not very familiar with the French healthcare system. Is it possible to trace the gastrointestinal conditions of these patients, such as unformed stools or other related issues?

Response: we agree that this is a preliminary study, which cannot aim to help the clinical practice.

Reviewer #2: The manuscript titled “Do Malassezia yeasts colonize the guts of people living with HIV?” from Stéphane Ranque’s laboratory presents analyses of stool samples collected from a longitudinal cohort. The study aims to compare the effects of HIV infection on Malassezia in human gut using conventional culture methods, metagenomics and PMA-qPCR. While the data presented are limited due to the small sample size, they are intriguing and could serve as a foundation for further research on this topic. However, I recommend the authors significantly improve the way they present the data before publication.

Major points: The primary issue lies in the presentation of the data, which is not easy to follow. Here, I will highlight some points that the authors may consider for improvement:

1. Title: While I like the title, I think that the manuscript did not fully address the question it poses. Since the data is not strong enough to provide a definitive answer, I suggest slightly modifying the title to better align with the study's findings.

Response: We appreciate that the reviewer likes the title. We acknowledge that our study does not provide a definitive answer to the question, so we prefer to keep the question mark. Still, we could not think of a better title.

2. Abstract: The authors may need to reorganize the abstract. It should be concise, with fewer methodological details and a stronger focus on the findings and conclusions. Readers will likely expect a more engaging abstract based on the manuscript’s title. As it stands, I lost motivation because the abstract does not clearly address the question posed in the title.

Response: We have rewritten the abstract in line with the reviewers' comments.

3. Main results: This section faces similar issues as the abstract. For each subsection, it would be more effective to present the key findings in the subsection titles rather than describing the results as if outlining research steps. Readers are more interested in what the data tells, not just what the data is or what the authors did. Due to this presentation style, the primary focus on Malassezia is not clearly emphasized.

Response: We have rewritten the Results section in line with the reviewers' comments.

4. Figure 1: Author could include a cartoon or schematic to illustrate your sample collection and patient information. This will give the audience an overview of the patient samples, including how many were used for culture, sequencing, how many patients completed the full study, and how many dropped out.

Response: We agree that a schematic illustrating sample collection and study flow would be helpful. However, due to the general disorganisation of patient care during the COVID-19 pandemic, minor protocol deviations make it difficult to draw a clear and simple scheme, and we would be reluctant to do so.

In Figure 1A, it would be helpful to include the number of positive cultures for each species. Additionally, labeling "Visit 1" and "All visits" more clearly in Figures 1A and 1B would improve clarity.

The font styles and sizes are inconsistent across the figures. Please use a consistent Sans-serif font, and increase the font size to improve visibility.

Response: The figure has been modified in accordance with the reviewers' comments.

There are many following figures based on only “Visit 1”. Authors can consider collecting them together in a big figure or indicate “Visit 1 analysis…” directly on these figures that would be easier to understand.

Response: For clarification, “Visit 1” has been added to each relevant Figure caption.

5. Figure S1: author may replace this figure with a table or bar plots showing the Ct values. The current figure is difficult to interpret due to the colors and lines, and it doesn’t provide much information. A table or bar plots would make it easier for readers to compare the data. The figure primarily needs to indicate which samples are PCR-positive (+) and which are negative (-). The current presentation is unnecessarily complex and takes too much time for readers to grasp the simple results.

Response: We agree that this figure is complex, which is why we have chosen to position it as a supplementary figure. However, we do not want to oversimplify our data by summarising it as positive or negative results in a table.

6. Figure 2: Although Figure 2B shows two significant comparisons, the Shannon diversity analysis may be biased due to the small sample size. Therefore, the figure essentially indicates no significant difference between the samples. The title of the figure should reflect this main result. Additionally, "Visit 1" should be indicated earlier in the title or directly on the graph for better clarity and understanding.

Response: We agree that the small sample size results in too little power to detect a statistically significant difference. However, we are required to present the results of our statistical tests, although we acknowledge this limitation in the discussion section.

7. Figure 4: The font size is too small and needs to be increased for better readability. Additionally, the organization of this figure could be improved. I suggest arranging the x-axis based on health conditions, from healthy to severe disease. This would make it easier to compare how the microbiome changes across different health conditions.

Response: We amended the figure for clarification.

8. Table 2: The table lacks meaningful impact unless the authors consider how many samples are positive and the variability of read counts between samples.

Response: There is a striking difference between the relative numbers in PLWH and controls, despite the variability in Malassezia sp. read counts.

9. Overall, there are so many descriptions, statistical number (percentage) that describe in the main text, however, lack of conclusion from each section and even each paragraph make the manuscript very hard to follow. I could not really catch the main finding of each section. Notably, the focus in this manuscript should be answering “Do Malassezia colonies the guts of HIV patients?” which are blurry because of many details and descriptions about statistical numbers of microbiome.

Response: We hope to give a clear answer to this question in the conclusion section, without being too conclusive due to the preliminary nature of our results. However, this study has generated a lot of data that cannot be found elsewhere. For this reason, we would like to present a detailed summary of our findings. We recognise that this will make the paper more difficult to read, but we think that some of these details may be of interest to the reader.

Therefore, while the data is valuable, the authors need to better organize the manuscript and focus on presenting the findings rather than just describing the data. Detailed descriptions should primarily be presented in the Figures and Tables, while the Results section should briefly summarize, analyze, and provide clear conclusions. The Discussion section is in good shape for now.

Response: We have done our best to follow the reviewer's recommendations. We acknowledge that he thinks the discussion section is in good shape.

Minor points:

1. Citations should be added before the sentence-ending period (".").

Response: this has been corrected.

2. Line 112: Please specify the volume used for plating on each plate.

Response: 100 µl, this has been detail in the methods section.

3. Will the metagenomic sequencing data be made publicly available? If there are no restrictions from the ethical agreement, the authors should consider depositing the data in a public repository.

Response: The data is available on https://www.mediterranee-infection.com/acces-ressources/donnees-pour-articles/data-gut-hiv/

4. Line 210: p value < 0.05

Response: corrected

5. Line 227: Wee

Response: We did not find this typo in the manuscript, hopefully we have corrected it.

6. Line 257: This is the first appearance of the genus “G.”.

Response: we spelled out the first appearance of the Geotrichum genus in the manuscript.

7. Line 345: Aspergillaceae is a family, not a genus.

Response: we corrected this typo in Table 3 and in the manuscript.

8. The authors should abbreviate terms such as "HIV-immunocompromised patients," "non-immunocompromised," "non-immunocompromised patients living with HIV," "immunocompromised and non-immunocompromised patients living with HIV," "CD4 < 200," and "CD4 > 500" for consistency and easier comparison of data. These abbreviations should be used consistently throughout the manuscript and figures to make the content easier to follow for a broad readership.

Response: We have abbreviated "person living with HIV" as "PLWH" and "immunocompromised" as "IC" throughout the manuscript.

9. Table 3: please check alignment of some lines.

Response: Alignment has been corrected.

6. PLOS authors have the option to publish the peer review history of their article (what does this mean?). If published, this will include your full peer review and any attached files.

Response: I agree to publish the peer-review history.

---

## [Decision Letter · Decision Letter 1]

1 Apr 2025

Do Malassezia yeasts colonize the guts of people living with HIV?

PONE-D-24-36362R1

Dear Dr. Ranque,

We’re pleased to inform you that your manuscript has been judged scientifically suitable for publication and will be formally accepted for publication once it meets all outstanding technical requirements.

Kind regards,

Kin Ming Tsui

Academic Editor

PLOS ONE

Additional Editor Comments (optional):

Thank you for the submission of a revised manuscript.

It could be acceptable for publication when the minor comments are addressed.

Reviewers' comments:

Reviewer's Responses to Questions

**Comments to the Author**

1. If the authors have adequately addressed your comments raised in a previous round of review and you feel that this manuscript is now acceptable for publication, you may indicate that here to bypass the “Comments to the Author” section, enter your conflict of interest statement in the “Confidential to Editor” section, and submit your "Accept" recommendation.

Reviewer #2: All comments have been addressed

2. Is the manuscript technically sound, and do the data support the conclusions?

Reviewer #2: Yes

3. Has the statistical analysis been performed appropriately and rigorously? 

Reviewer #2: (No Response)

4. Have the authors made all data underlying the findings in their manuscript fully available?

Reviewer #2: No

5. Is the manuscript presented in an intelligible fashion and written in standard English?

Reviewer #2: (No Response)

6. Review Comments to the Author

Reviewer #2: 1. Please revise Table 3 and other relevant sections: use "spp." (for multiple species) instead of "sp.", which refers to a single species.

2. In Figure 2, the result shows P < .05, whereas in Figure S2, p > 0.05 — please ensure a consistent format.

3. Raw sequencing data must be deposited in a public repository such as NCBI GenBank Metagenome https://www.ncbi.nlm.nih.gov/genbank/metagenome/. Processed data should be included as supplementary material, not uploaded as an Excel file to a institute website.

7. PLOS authors have the option to publish the peer review history of their article (what does this mean? ). If published, this will include your full peer review and any attached files.

**Do you want your identity to be public for this peer review?** For information about this choice, including consent withdrawal, please see our Privacy Policy .

Reviewer #2: No

---

## [Editor Report · Acceptance letter]

PONE-D-24-36362R1

PLOS ONE

Dear Dr. Ranque,

I'm pleased to inform you that your manuscript has been deemed suitable for publication in PLOS ONE. Congratulations! Your manuscript is now being handed over to our production team.

Kind regards,

on behalf of

Dr. Kin Ming Tsui

Academic Editor

PLOS ONE